# The Downside of Heterogeneity: How Established Relations Counteract Systemic Adaptivity in Tasks Assignments

**DOI:** 10.3390/e23121677

**Published:** 2021-12-14

**Authors:** Giona Casiraghi, Christian Zingg, Frank Schweitzer

**Affiliations:** Department of Management, Technology, and Economics, ETH Zürich, Weinbergstrasse 56/58, 8092 Zürich, Switzerland; czingg@ethz.ch (C.Z.); fschweitzer@ethz.ch (F.S.)

**Keywords:** resilience, systemic risk, failure cascades, entropy, adaptivity

## Abstract

We study the lock-in effect in a network of task assignments. Agents have a heterogeneous fitness for solving tasks and can redistribute unfinished tasks to other agents. They learn over time to whom to reassign tasks and preferably choose agents with higher fitness. A lock-in occurs if reassignments can no longer adapt. Agents overwhelmed with tasks then fail, leading to failure cascades. We find that the probability for lock-ins and systemic failures increase with the heterogeneity in fitness values. To study this dependence, we use the Shannon entropy of the network of task assignments. A detailed discussion links our findings to the problem of resilience and observations in social systems.

## 1. Introduction

Imagine a world in which every task is solved right when it appears. Agents with huge mental capacities and time resources would immediately execute whatever was assigned to them. No unfinished tasks would pile up at their desks, no procrastination, no delay in project management. A measure of systemic performance that monitors the concentration of unfinished tasks would always be at its optimum.

Unfortunately, the real world is not like this, on the contrary. Miraculously, the work to be done seems to pile up at the desks of some agents, bottlenecks emerge. Zanetti et al. [1] show that these bottlenecks are not necessarily created by the least productive agents but, in some circumstances, by the most productive ones. The reason for this miracle comes from the underlying feedback. Agents that have proven to solve tasks assigned to them very quickly in the past have a larger chance to get new tasks assigned [2]. Therefore, the system “learns” to preferably consider those agents, this way overwhelming them with new tasks. Surely, these overcommitted agents could redistribute such tasks themselves to others. However, the problem is that those agents receiving the task may not be more efficient and simply pass the assignment on to others. Therefore, instead of solving the task, a cascade of subsequent assignments emerges. In the worst case, every agent redistributes all tasks, resulting in a system where no task is solved anymore [3].

This quite abstract problem description usually resonates with the personal experience of a broader audience, which has experienced similar situations already. Therefore, we do not further illustrate the problem but go straight to the research question: Can a lack of adaptivity in redistributing tasks lead to a system failure under certain conditions? More precisely, we are interested in distinguishing dynamic regimes where the system can still maintain its task solving ability from those where the system breaks down. This would require us to also monitor the systemic state by a suitable measure—which turns out to be Shannon’s information entropy [4] in our case.

In the following, we introduce the details of our agent-based model of task redistribution. By means of computer simulations we find that the risk of failure cascades increases with the heterogeneity in the agents’ fitness, i.e., their ability to solve tasks. This results from the fact that agents learn to redistribute their unfinished tasks to those agents with a higher fitness, which then fail. Our findings are related to observations in an open source software project. We further discuss their relevance for understanding systemic risk and resilience.

## 2. Agent-Based Model of Task Redistribution

Agent fitness

In our agent-based model, each agent i=1,…,N is characterised by a dynamic variable $x_{i}\left(t\right)$, giving the number of tasks agent *i* still has to solve at time *t* in a continuous approximation. A task is a discrete unit, but an agent solves it over a period of time, which means that the agent solves a fraction of the remaining work at every *t*. Each agent *i* is described by two time-independent attributes that quantify its ability to process tasks: its fitness $\phi_{i}\in R^{+}$ and its performance $\tau_{i}\in \left[0,1\right]$. The performance $\tau_{i}$ denotes the *rate* at which an agent *i* solves tasks. The larger $\tau_{i}$, the faster an individual solves tasks. We discuss in details the role of $\tau_{i}$ below. The fitness $\phi_{i}$, instead, can be seen as some sort of individual capacity to handle tasks and defines the *heterogeneity* of agents. Agents with $\phi_{i}=0$ cannot handle any task, while agents with $\phi_{i}=\infty$ can handle any number of tasks.

Agent failure

Agents will remain *active* as long as $x_{i}\left(t\right)\leq \phi_{i}$, i.e.,as long as they can still handle tasks assigned to them. If $x_{i}\left(t\right)>\phi_{i}$, agents *fail* and become inactive. This is described by an individual state variable [5]:(1)$$\begin{array}{c} s_{i}\left(t\right)=\Theta \left[x_{i}\left(t\right)-\phi_{i}\right] \end{array}$$
where Θ[z] is the Heaviside function, which returns 1 if z≥0 and 0 otherwise. Note that, according to Equation (Equation 1), for an agent with fitness $\phi_{i}\leq 0$ we have that $s_{i}\left(t\right)=1$ for all t≥0. The state variable $s_{i}$ can be used to calculate the fraction of failed agents as follows:(2)$$\begin{array}{c} X\left(t\right)=\frac{1}{N}\sum_{i}s_{i}\left(t\right) \end{array}$$
X(t) can be used as a measure of *systemic risk* [5,6]. In that framework $x_{i}\left(t\right)$ is denoted as susceptibility, reflecting the internal and external influences on the agent performance, while $\phi_{i}$ is a threshold that expresses the level of internal stability, or “healthiness”, of an agent. We will link our discussion back to issues of systemic risk at the end of this paper.

Obviously, the choice of $\phi_{i}$ to a large extent determines the success and failure on the systemic level. To assign individual fitness values to agents, we sample them from a normal distribution N(μ,σ), where μ=50 is the mean value and σ is the standard deviation of the distribution. In the following, we will vary σ, i.e.,we increase or decrease the heterogeneity of agents. We then study the impact of the heterogeneity σ on the system’s state in Section 3.

Task increase and decrease

Initially, each agent gets assigned a number of tasks $x_{i}\left(0\right)=15$ at t=0. As long as an agent is active, $x_{i}\left(t\right)\leq \phi_{i}$, it can solve these tasks at the rate $\tau_{i}$. In a continuous time approximation the dynamics then reads [7]
(3)$$\frac{dx_{i}\left(t\right)}{dt}=-\tau_{i}x_{i}\left(t\right)$$

This implies that tasks can be solved in parallel, and fractions of tasks remain. Because an agent will solve tasks faster if it receives more tasks, the amount of unsolved tasks decays exponentially. In this paper, we are interested in the impact of *fitness* heterogeneity. Therefore, we set an equal performance for all agents, $\tau_{i}=\tau =0.01$, in the simulations described below.

Tasks are not only solved. Each active agent also receives new tasks based on two processes. The first one assumes the arrival of new tasks on each agent’s desk at constant time intervals. The second one assumes a *redistribution* of unfinished tasks between agents. To model these two processes, we consider a second discrete time scale *T*. Hence, our multi-agent system is characterised by *two time-scales*, a shorter continuous one and a larger discrete one. At the shorter time scale, *t*, agents process the assigned tasks, while at the larger time scale *T*, the arrival of new tasks and the redistribution of unfinished ones occur.

Here, we consider that an agent can only redistribute *full tasks*. This is expressed by the floor function, $\left\lfloor x_{i}\left(T\right)\right\rfloor$, which separates the full tasks that can be distributed from those tasks that the agent has already started to solve and that cannot be distributed anymore. Precisely, at every time step *T*, agent *i* chooses to reassign some of its $\left\lfloor x_{i}\left(T\right)\right\rfloor$ uncompleted tasks to other agents with a probability:(4)$$p_{i}\left(T\right)=\frac{\left\lfloor x_{i}\left(T\right)\right\rfloor }{\phi_{i}}.$$

This probability increases with the workload of agent *i*, scaled by its ability to solve tasks. When a task is reassigned, $\left\lfloor x_{i}\left(T\right)\right\rfloor$ and $p_{i}\left(T\right)$ are updated instantaneously, i.e.,they decrease with each task reassigned. With this assumption, we can describe the process of choosing which tasks are reassigned at every discrete time step *T* as a sampling without replacement. From an urn that contains exactly $\left\lfloor x_{i}\left(T\right)\right\rfloor$ green balls (which can be redistributed), and $\phi_{i}-\left\lfloor x_{i}\left(T\right)\right\rfloor$ red balls (which cannot be redistributed), we sample $\left\lfloor x_{i}\left(T\right)\right\rfloor$ balls. The number of green balls in this sample gives the number $D_{i}\left(T\right)$ of tasks to redistribute. Under these conditions, the total number of tasks to be redistributed $D_{i}\left(T\right)$ follows a hypergeometric distribution. Thus, after every discrete time step *T*, the number of tasks to solve drops according to the following equation:(5)$$x_{i}\left(T+\varepsilon \right)=x_{i}\left(T\right)-D_{i}\left(T\right),$$
where ε→0.

Task redistribution

To specify how agents redistribute their tasks to others, we make two assumptions. First, agent *i* reassigns tasks to other agents *j* proportional to their fitness $\phi_{j}$. This implies that agents know all fitness values and can decide freely to whom to reassign their tasks. Second, agents learn to whom they reassign tasks: the more an agent was chosen in the past, the more likely it will be chosen again. A counter $w_{ij}\left(t-\varepsilon \right)$ gives the number of times *i* has reassigned a task to *j* in the past. Again, we assume that tasks are distributed instantaneously and that $w_{ij}$ is updated accordingly.

Hence, from the $D_{i}\left(T\right)$ full tasks that have to be redistributed, each one is reassigned to an agent j≠i with probability $q_{ij}$:(6)$$q_{ij}\left(t\right)∼\phi_{j}\cdot \left[w_{ij}\left(t-\varepsilon \right)+1\right]$$

If all $\phi_{j}=\phi$ are constant, the reassignment dynamics would be described by a simple multivariate Polya urn process, distributed according to the standard Dirichlet-multinomial distribution [8]. For heterogeneous $\phi_{i}$, however, the dynamics is described by a generalised multivariate Polya urn process and follows a form of modified Dirichlet-multinomial distribution [9]. Investigating the exact form of this distribution is beyond the scope of this paper.

Full dynamics

If agent *i* redistributes $D_{i}\left(T\right)$ full tasks at time *T*, then $d_{ij}\left(T\right)$ is the number of tasks that agent *i* assigns to *j* at time *T*. At the same time, agent *i* may receive tasks assigned from other agents, and at constant time intervals $T_{new}=10$ one new task arrives. Thus, for the number of tasks $x_{i}\left(t\right)$ of agent *i*, we eventually specify the full dynamics:(7)$$\begin{array}{cc} \frac{dx_{i}\left(t\right)}{dt}= & -\tau \left[x_{i}\left(t\right)-\delta_{t\in N}\cdot D_{i}\left(t\right)\right]-\delta_{t\in N}\cdot D_{i}\left(t\right)\\ \; & +\delta_{t\in N}\cdot \sum_{j}d_{ji}\left(t\right)+\delta_{t\;\mathrm{mod}T_{\mathrm{new}}=0} \end{array}$$

This dynamics merges the different processes at the continuous time scale $t\in R^{+}$ and the discrete time scale T∈N by using the expressions $\delta_{a=b}$. They denote the Dirac delta that is 1 when a=b and 0 otherwise. If $x_{i}\left(t\right)\geq \phi_{i}$, agent *i* fails and redistributes all of its tasks to others. An agent who fails can no longer receive or solve any tasks in the future.

Network of reassignments

The task redistribution dynamics generates a network of reassignments between all agents. Specific paths i→j→⋯→k in this network can be reinforced over time because of the memory effect: agents choose previously chosen agents with a higher probability. Depending on the parameter values for this model, this can lead to lock-in effects. I.e., agents are no longer flexible enough to choose different agents for their reassignment. At the systemic level, this results in a loss of adaptivity. Thus, the efficient redistribution of tasks is hampered. This outcome can even lead to a failure of the system. If too many agents are overwhelmed with tasks above their capacity, i.e.,$x_{i}\left(t\right)\geq \phi_{i}$, they fail and redistribute all their tasks to the remaining active agents, possibly triggering a cascade of failures.

Hence, we can describe the systemic dynamics by monitoring the network of reassignments over time. The nodes of this network represent the active agents, and the weighted and directed links result from the reassignments. Because agents can freely choose to whom they reassign tasks, the network can become fully connected over time. The weights $w_{ij}\left(t\right)$ can be used to define the entries $A_{ij}\left(t\right)$ of an adjacency matrix A(t), that characterizes this network. These $A_{ij}\left(t\right)$ evolve over time, hence the network constantly adapts.

An illustration of this process is shown in Figure 1. We note that major changes in the network occur only during the first few time steps. Then the network “locks in” to a quasistationary state that cannot easily adapt. In the end, tasks are only solved by a small number of agents that survived.

## 3. Results of Agent-Based Simulations

### 3.1. Evolution of Task Reassignments

Figure 1 has shown a large heterogeneity of the network, both in terms of the number of tasks agents have to solve (size of the nodes) and the number of tasks redistributed (size and color of links). The latter is captured in the different $A_{ij}\left(t\right)$ of the adjacency matrix A. Hence, we propose to utilize this information for a *systemic measure* that reflects the evolution of task reassignment. This measure is Shannon’s *information entropy* [4], defined as:(8)$$H\left(A\right):=-\sum_{A_{ij}\in A}\frac{A_{ij}}{\sum_{A_{ij}\in A}}\cdot \log_{2}\left(\frac{A_{ij}}{\sum_{A_{ij}\in A}}\right),$$
H(A) quantifies the heterogeneity of entries in the matrix. The maximum value for H(A) is attained when all entries are identical. For a system of *N* agents, the theoretical maximum value is given by
(9)$$\max\left\{H(A)\right\}:=\frac{N(N-1)}{N^{2}}\log_{2}\left(N^{2}\right),$$
where the term N(N−1) comes from the fact that the diagonal of the adjacency matrix is always zero because an agent does not redistribute tasks to itself. Conversely, the more heterogeneous the entries, the lower is the value of the corresponding information entropy H(A).

Specifically, monitoring the *dynamics* of H(A(t)) allows us to characterize the extent to which the system is in a lock-in state [10]. This state is reached if the interactions among agents follow a fixed pattern, e.g., reassigned tasks will be always sent to the same agent. The lower the information entropy, the higher is the degree of lock-in.

In Figure 2 we show the dynamics of the Shannon entropy for three exemplary parameter values of σ. The corresponding network structures are shown in the lower part of Figure 2. For σ=0 we quickly observe an approximately stationary state because the system does *not* lock-in. H(A)≈6 is the maximum value for the given choice of parameters. Because all agents have the same fitness, they eventually contribute equally to solving the tasks. However, as the assignment network below for an intermediate time indicates, the structure of the reassignments first has to be established. For σ=9, we observe a slow decrease in the Shannon entropy with little variation, after it has reached its largest value close to the theoretical maximum. This corresponds to a partial lock-in of the system. We note that in this case the network is rather sparse, indicating a loss of adaptivity in redistributing tasks.

The most interesting plot is the one for σ=16. Different from the other plots, we now see three very different curves for the entropy. They correspond to three possible outcomes that can happen only for large heterogeneity. The most frequently observed green curve again shows the considerable decrease of the Shannon entropy, after it has reached its largest value. This indicates a partial lock-in, but stronger than for smaller σ. *Additionally*, the violet curve illustrates the scenario of a *complete failure*. After the initial maximum, the entropy sharply drops down to reach very low values. In this situation, some agents become overwhelmed and fail, and therefore, their unfinished tasks have to be redistributed to the remaining agents. This partially improves the situation: Even with fewer agents, the task reassignment becomes better balanced, shown in the increasing entropy. However after some time, other agents fail, which leads again to a redistribution and an entropy increase. Eventually, the few remaining agents cannot handle all tasks and also fail. At this point, the violet curve stops, because no agents are active anymore. This can be seen as a *failure cascade* [5] that encompasses the whole system in the end. However we emphasise that this cascade happens over quite a long time period, so it cannot be simply reduced to a domino effect with immediate impact on other agents.

Additionally, the orange curve also leads to a steep decrease of the entropy, but then illustrates a very different scenario. After the failure of some agents, the system finds a better balance for redistributing the tasks, which even improves further over time. Hence, compared to the two other curves, a sustainable and well balanced quasistationary state has been obtained. We will continue this discussion in Section 4.3 when we refer to the notion of *resilience*.

### 3.2. Impact of Heterogeneity

To further characterize the state of the system over time, we use the measure of systemic risk, X(t), Equation (Equation 2), which gives the fraction of failed agents. Values X→1 for long times indicate that the system has broken down. We are interested in studying how this failure depends on the *heterogeneity* of the agent fitness, expressed by σ. Figure 3 shows that for small σ the fraction of failed agents stays close to 0, i.e.,the majority of agents survives. However this fraction considerably increases if the heterogeneity becomes larger, which demonstrates the *negative* role of σ for the survival of the system.

Remarkably, for large σ the fraction of failed agents follows a bimodal distribution (cf. inset in Figure 3). This means for large σ
*either* only a few agents fail *or* almost all of them do. Partial losses between 50% and 85% almost never occur because the failure cascades, and if they occur, in case of large heterogeneity, they become also large Note that, for large values of σ there is a non-zero probability to observe fraction of failed agents larger that 0 at t=0. However, this does not impact the results discussed here as these failed agents do not considerably impact the dynamics.. Hence, an increase of heterogeneity in the agents’ fitness values $\phi_{i}$ not only leads to a complete lock-in of the system, but also has the risk of a complete system failure.

To further study the role of the heterogeneity parameter σ, in Figure 4 we plot a heat-map of the average Shannon entropy over time. The darker the colour, the lower the average value of the entropy, that is, the stronger the lock-in effect in the redistribution network. We identify a threshold value around σ = 5, above which a system failure starts to be observed. Below the threshold, the system is either in partial lock-in or does not reach the lock-in at all. Above the threshold, i.e., for higher values of σ, the system reaches complete lock-in, followed by the failure of agents. This happens the earlier the higher the values of σ are.

## 4. Discussion

### 4.1. A Realistic Example

In this paper, we have provided a model of task reassignment. The underlying feedback mechanism is *learning*: agents learn to reassign tasks they are not able or willing to solve to other agents with higher fitness. Our model does not consider that those reassignments can be rejected. Instead, agents can again forward these tasks to others if they are not processed. If agents get too many tasks, they will eventually fail. The probability for reassignment decreases with the agent’s fitness. Hence, in the end, most tasks will pile up at the desks of those agents with the highest fitness.

We first want to address the question to what extent this is a realistic scenario. Here, we refer to a case study by Zanetti et al. [1] about task assignments in the large-scale Open Source Software (OSS) project Gentoo. Specialised communities for tasks like bug handling are of particular importance for the success of such projects [11]. Therefore, the management has to find efficient organisational structures for the division of labour [12,13], even though these communities are, typically, highly heterogeneous in dedication and skills [14]. Some developers only contribute a single time, while core-developers perform the majority of work [15]. From an extensive analysis of the network of task assignments, Zanetti et al. [16] found that over time developers in Gentoo tended to rely mainly on a single central contributor who became responsible for handling most of the tasks. This concentration, however, considerably reduced the *resilience* of the system against shocks. Because everything depended on *one* central contributor, there was no redundancy in the ability to handle tasks. This lack of redundancy exposed the whole project to a considerable failure risk if this central contributor dropped out [17]. This did indeed happen when the central contributor became overwhelmed by tasks and faced conflicts originating from the task reassignment [1].

In Figure 5, we show the network of task assignments at different stages of the Gentoo project. In an early phase, tasks were broadly distributed between a core of active developers and a large periphery of less active ones. As the number of tasks has grown considerably, in the second stage, we see that the system almost dissolves into two groups, indicated by the two large network components. These components are mainly connected only by the central contributor which makes the system prone to failure because the star-like network structure [1]. Thus, after the central contributor left, the system broke down and had to reorganise. The result of this adaptation process is shown in the third phase. The most important node now represents a *software*, Bugwrangler. It helped achieve a much more balanced task assignment, and a single contributor no longer dominates the system.

From this real-world case study, our model can capture the concentration of tasks in very few individuals, which are the ones with the highest fitness, and the resulting breakdown after they become overloaded. Hence, our model can shed new light on the conditions under which such vulnerable states appear. Based on our modelling insights, we argue that the Gentoo project described in [1] suffered from a lock-in effect that led to its breakdown. Lock-in means that the redistribution of tasks was constrained by its previous history, dominated by the activities of the central contributor. Therefore, the system could not respond to necessary changes anymore. The recovery process became only possible after the central contributor left and most of the established relations broke down. While we have seen that our model also allows for scenarios of obtaining a much better redistribution balance (see Figure 2 right), we do not claim that the recovery of the Gentoo project is sufficiently captured. For Gentoo, the last stage of reorganization and adaptation shown in Figure 5c also relied on additional resources (manpower, software).

### 4.2. Systemic Risk

The model we have proposed has much in common with other models of systemic risk, but also some remarkable differences that we want to discuss now. Systemic risk denotes the risk that a large part of the system fails [5]. It can occur because of extreme shocks that destroy the system but also because the failure of a few agents is amplified [18,19,20]. To understand the latter case, one has to move from estimating the probabilities of rare events to modelling (i) agents’ *interactions* and (ii) their *internal dynamics*. Agent-based models are best suited for that, in particular in combination with network models [7].

Our model falls into the class of *load redistribution* models [5,6,21,22]. They assume that a failing agent redistributes its load, i.e.,the unfinished tasks, equally to its neighbours. Because their load increases, these agents have a higher probability of failing. Hence, a failure cascade emerges, which could stop after some steps, but more often accelerates. This is because redistribution models capture a double amplification: more and more agents fail and distribute their load to fewer and fewer agents that are still active [23].

Our model deviates from this outline in different respects.

In our case, agents *continuously* redistribute tasks, not only if they fail.In our model, agents do not *equally* redistribute their tasks. Instead, the number of tasks redistributed at each time step follows a probability distribution that combines agent features, i.e.,all their fitness values and the history of previous reassignments.Our model considers *directed* and *weighted* links, where the weight dynamically adjusts according to the history of assignments. i.e.,instead of a static network topology, our model uses an adaptive network [24] which reflects the learning process of agents.As our model simulations illustrate, the failure of some agents not necessarily worsens the situation but sometimes leads to a better redistribution of tasks, i.e.,to an improved system state. This is not reflected in most redistribution models, with the fibre bundle model [25,26] as a paragon, because only failing agents redistribute their load, negatively impacting the stability of the system.

One main finding from our simulations is the *bimodal* distribution for the fraction of failed agents (Figure 3). That means, in addition to the existence of small failure cascades, there is a considerable risk that the *whole system* fails under the very *same conditions*. This finding is at odds with known results for infinitely large systems, where a unimodal distribution was obtained [22]. In this case the *average size* of a failure cascade is a good predictor for the systemic risk. For a bimodal distribution, however, the average cascade size is precisely *not* representative for the system behaviour. We have already explored that the *finite system size* is the reason for the bimodal distribution [23]. This makes our findings even more relevant as real systems are always finite. Therefore, we can rightly expect that our system of task redistribution behaves differently from a theoretical limit case, also with respect to the risk of a complete failure.

### 4.3. Conditions for Resilience

The focus of our paper is the impact of agents’ heterogeneity on the ability of the system to adapt. The main finding states that a larger heterogeneity in the fitness distribution results in more substantial lock-in effects. In the case of low heterogeneity, however, lock-ins do not occur. This finding is not trivial because the learning dynamics still tends to drive the system into a lock-in state. Nevertheless, this effect is counterbalanced by the fact that agents have more choices to redistribute their tasks because all agents have the same fitness. These counterbalancing effects result in a higher adaptivity in the end.

Lock-in means that agents always reassign their tasks to the same agents as before. Therefore, the reassignment network becomes very sparse, and the system loses its ability to adapt. As we have shown, this does not mean the system will break down. However a strong lock-in effect increases the chances that agents fail, which in turn increases the chances that failure cascades evolve.

A system is said to be *resilient* if it can adapt to shocks and even recover from them [27,28,29]. Thus, a loss of adaptivity undoubtedly impacts resilience, but it does not explain why the system breaks down in some cases and in others not. In Figure 2c, we have plotted three curves from different simulations with large heterogeneity. The green curve shows the typical behaviour: agents learn to whom they reassign their tasks. The lock-in is the result of this adaptation on the systemic level. There are no “shocks”, and there is no recovery.

The violet and the orange curve, on the other hand, display such shocks: agents fail, and the system has to respond by additionally redistributing their tasks. The failure of agents first results in a low adaptivity of the system. Remarkably, the drop-down in adaptivity is not caused by agents with low fitness, which may still be active, but by the loss of agents with high fitness, which cannot be easily replaced because the system has “learned” to rely on them. After this drop-down of adaptivity, we see a positive response for both the orange and the violet curves first: the entropy increases again, which means the system regains a better-balanced state, i.e., adapted to the shock. However, in the case of the violet curve, this ability could not ensure the long-term existence of the system. The explanation lies in the fact that *resilience* has *two* constituting components [30]: *adaptivity* only denotes the dynamic dimension [31,32], while *robustness* denotes the *structural* dimension [29,33]. As the difference between the orange and the violet curve makes clear, an increase of adaptivity alone is not sufficient [34,35]. There has to be also an increase in robustness. This was the case for the scenario pictured in the orange curve. Precisely, the system not only recovered from the shock but even obtained a balanced state better than most of the typical simulations—and kept this over a long time.

Eventually, we note one of the most important differences to other models of risk and resilience. In most systems, the failure cascade that characterises the breakdown starts from agents with the smallest threshold, a measure of their “healthiness” or fitness. This is understandable because they are “weak” and cannot handle a larger load. In our case, however, the failure cascade always starts from agents with higher fitness, considered as “strong”. This seems to be counterintuitive but is in line with our real-world example described in Section 4.1. The reason for this observation is the existing positive feedback: *Because* other agents are perceived as “strong”, they get assigned most of the work. This process is not counterbalanced by negative feedbacks that would stabilise the system, e.g. through saturation or competition. Hence, tasks can pile up to a level where even the strongest agents get overwhelmed and fail. The weaker agents, on the other hand, are better protected because of their larger reassignment rate.

This links our investigation to other studies [36,37] that show how protecting a periphery of low performing agents can decrease the robustness of a system. In conclusion, protecting the weakest agents does not prevent the breakdown—on the contrary, it enables systemic failure by sacrificing the strongest instead [36,37]. However, without the stronger agents, a recovery will become even more difficult because it can solely rely on the weak agents [30]. Such insights, derived from a rather abstract model, have the potential to let us rethink the distribution of tasks and resources in entirely unrelated systems. Protecting the weak is a good idea, but to achieve an increase in resilience, it would be better to understand the unintended consequences.

## Figures and Tables

**Figure 1 entropy-23-01677-f001:**
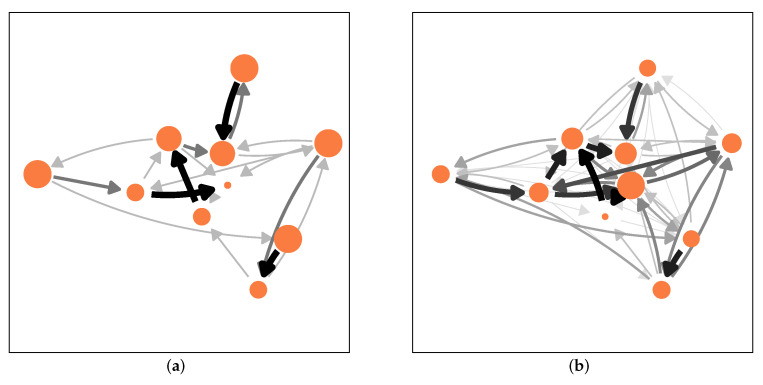
Network of task reassignments at different time steps: (**a**) *t* = 1, (**b**) *t* = 5, (**c**) *t* = 10, (**d**) *t* = 300. The size of the nodes is proportional to the number of tasks $x_{i}\left(t\right)$ assigned to them, the size and gray scale of links is proportional to the flow of tasks. Nodes in grey have failed. Parameters: *N* = 10, σ=8.5.

**Figure 2 entropy-23-01677-f002:**
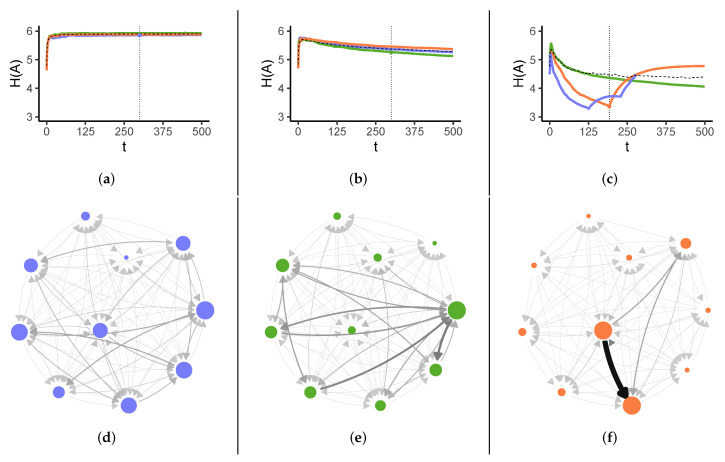
(**top row**) Shannon Entropy H(A(t)) over time *t* for different σ: (**left**) σ = 0, (**middle**) σ = 9, (**right**) σ = 16. The different colours are for three different runs, dashed lines are averages over 500 independent simulations. (**bottom row**) Reassignment networks from the respective curves above, where *t* is indicated by the vertical dotted line.

**Figure 3 entropy-23-01677-f003:**
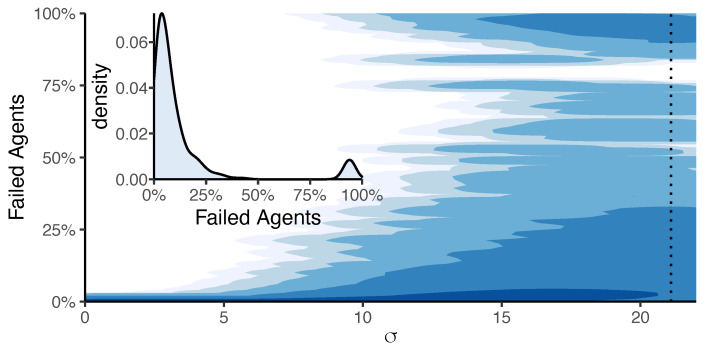
The heatmap shows the estimated probability density of X(t) (obtained from a standard kernel density estimation) dependent on σ. White corresponds to values of approximately 0, the darker the color, the higher the density. Parameters: *N* = 100. Values are averaged over 500 simulations of 1000 time steps for each value of σ linearly spaced in [0,22] (25,000,000 data points). Inset: Bimodal probability density estimate of X(t) for the σ value indicated by the dotted line.

**Figure 4 entropy-23-01677-f004:**
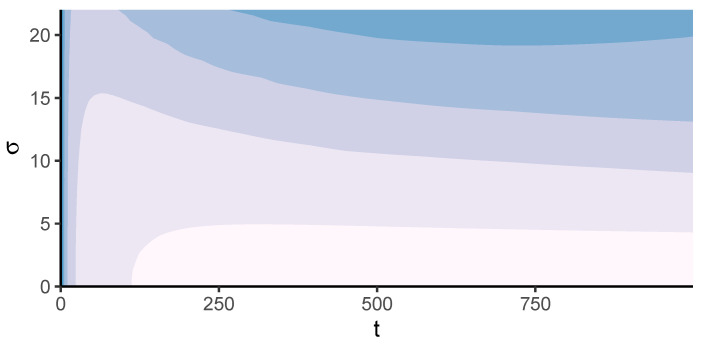
Heat-map of the average Shannon entropy over time for different values of σ. Light colors correspond to high entropy values, close to the theoretical maximum. dark colors to locked-in states. Values are averaged over 500 simulations for values of σ between 0 and 22. Simulations run until all agents fail or stop redistributing tasks.

**Figure 5 entropy-23-01677-f005:**
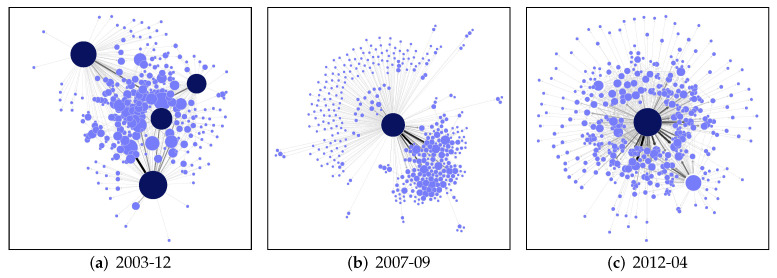
Snapshots of a partial developer network from the OSS project Gentoo. Data is redrawn from [1]. The size of the nodes is proportional to the number of tasks assigned.

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
