# Peer review of "The Downside of Heterogeneity: How Established Relations Counteract Systemic Adaptivity in Tasks Assignments"

_entropy, 2021, doi:10.3390/e23121677_

Round 1
Reviewer 1 Report
The paper offers a model of redistribution of workload, presumably within an organization
or an enterprise. A new element is that tasks are directed mainly to more efficient agents,
which got more tasks already in the past. Main result is that a failure of the system is
related to this particular rule.
The quality of the research is somewhat reduced by some model assumptions, which are far
from realistic. In particular, agents which failed from overload are eliminated from the
system (page 4, line 96). This assumption clearly magnifies the risk of failure.
Therefore the accompanying discussion, altogether with the comparison with literature (case
of GENTOO project) suggests a closer relation of the model with reality than it actually is.
However, I fully agree with the conclusion on the role of redistribution of tasks between
strong/weak agents.
The presentation is perfect. My only doubt is about eq 3: as the parameter \tau_i is positive,
x_i(t) exponentially increases. However, x_i(t) is a number of tasks, which is supposed to
decay exponentially (page 2, line 66). Then I would expect a minus sign on the right hand
side of eq. 3.
On the other hand, the statement about experience (page 1, line 17) seems premature. I would
suggest to add a reference to this statement, as it is ahead of the main goal presented
in the paper.
With these corrections, I recommend publication.
Author Response
We thank the reviewer for the constructive feedback and the comments provided.
Below, we provide a point by point response.
The quality of the research is somewhat reduced by some model assumptions, which are far
from realistic. In particular, agents which failed from overload are eliminated from the
system (page 4, line 96). This assumption clearly magnifies the risk of failure.
Therefore the accompanying discussion, altogether with the comparison with literature (case
of GENTOO project) suggests a closer relation of the model with reality than it actually is.
However, I fully agree with the conclusion on the role of redistribution of tasks between
strong/weak agents.
We agree with the reviewer that some of our model assumptions are not realistic.
However, we chose such simplifying assumptions to ease the understanding of the role of redistribution of tasks between agents and how this may impact the ability of the system to adapt.
By doing so, we can single out and study some aspects of real-world systems that would otherwise be impossible to disentangle if we were to use a more realistic and complex model.
The presentation is perfect. My only doubt is about eq 3: as the parameter \tau_i is positive,
x_i(t) exponentially increases. However, x_i(t) is a number of tasks, which is supposed to
decay exponentially (page 2, line 66). Then I would expect a minus sign on the right hand
side of eq. 3.
As noted also by the second reviewer, there is a typo in eq3. Indeed, there should by a minus sign before \tau_i.
We have corrected the equation in the revised manuscript.
On the other hand, the statement about experience (page 1, line 17) seems premature. I would
suggest to add a reference to this statement, as it is ahead of the main goal presented
in the paper.
We agree with the reviewer that the statement may be too strong.
To solve the issue, we have adapted the sentence and referred to the work of Zanetti et alii.
Reviewer 2 Report
The manuscript under review studies a model of reassignment of tasks in a complex population of interacting agents. Agents have to solve tasks and can redistribute their tasks to the more efficient ones with preference towards who has already taken more tasks in the past.
The paper is well written, clear and its contribution, though not revolutionary, is not simply incremental in its field. I support the publication, after the authors address few points I write down below.
1) The performance $\tau_i$ is introduced at line 48 (page 2), but explained explicitly only at line 64: I would suggest to write immediately that these taus are the rate of task solving;
2) About Equation (3), page 2: shouldn't be there a minus at right side? The more rapidly they solve tasks, the quicker should decrease the number of tasks $x(t)$, or have I missed something?
3) Equation (7), page 4: here we find correctly the term $-\tau x_i(t)$, but inside the parenthesis at right side I'd expect the term $+\delta_{t\in N} D_i(t)$, because again the more the agent $i$ redistributes tasks, the more $x_i(t)$ should decrease;
4) The distribution of fitness, $\phi_i$, is a Gaussian with $\mu=50$ and different standard deviations $\sigma$ (line 59, page 2): that means that there is always the possibility to assign negative values to the fitness. Now, for small values of $\sigma$ and small systems this probability is more than negligible, but, for instance, with $\sigma=22$ and $N=100$ (values utilized in Figure 3), it implies that on average 1.3 agents have a negative fitness for each realization, so that we probably in some realizations there were several agents with negative fitness. I do not think this changes apreciably the results obtained, but how did you treat these few cases? If negative, the fitness is like zero, but this means that the "real" distribution is not exactly a normal one. I understand that the practical effect is not important, but it would be better to clarify this point;
5) Figure 2, page 6: for the systems utilized, the maximum of $H(A$) is around 6, which implies that the number of agents is 20, but the system depicted are made up by 10 nodes (which would imply a maximum of $H(A)$ around 4). Are the networks depicted in the figure the real networks of the simulated systems?
Minor points:
6) Figure 3, page 7: the heatmap shows what is called the "probability" of $X(t)$ for given values of failed agents' ratio and $\sigma$, but I'd call it "frequency" (I do not mean "probability" is wrong, but since I presume it was measured just counting the realizations ended up with the given value of $X(t)$, "frequency" is in my opinion better);
7) Throughout the paper, there are figures with "red" curves, nodes, etc., but it seems orange (unless there is some problem with the visualization in my laptop). Of course it is not really an issue but just to be precise.
Apart the points raised above, I enjoyed the paper and after a minor revision to fix them I will endorse it for publication.
Author Response
The manuscript under review studies a model of reassignment of tasks in a complex population of interacting agents. Agents have to solve tasks and can redistribute their tasks to the more efficient ones with preference towards who has already taken more tasks in the past.
The paper is well written, clear and its contribution, though not revolutionary, is not simply incremental in its field. I support the publication, after the authors address few points I write down below.
We thank the reviewer for the positive assessment and the detailed comments. Below, we provide a point by point response.
1) The performance $\tau_i$ is introduced at line 48 (page 2), but explained explicitly only at line 64: I would suggest to write immediately that these taus are the rate of task solving;
We have expanded the description of $\tau_i$ as suggested by the reviewer.
2) About Equation (3), page 2: shouldn't be there a minus at right side? The more rapidly they solve tasks, the quicker should decrease the number of tasks $x(t)$, or have I missed something?
Correct. As mentioned in our answer to Reviewer 1, there is indeed a typo in eq3 which we have solved in the revised manuscript.
3) Equation (7), page 4: here we find correctly the term $-\tau x_i(t)$, but inside the parenthesis at right side I'd expect the term $+\delta_{t\in N} D_i(t)$, because again the more the agent $i$ redistributes tasks, the more $x_i(t)$ should decrease;
Thanks for pointing this out. Here too there is a typo.
The correct equation is, however, slightly different than the one suggested by the reviewer.
At every time step, we can separate the dynamics into two parts: (A) Tasks that are solved, and (B) tasks that are distributed or received.
According to eq5, tasks to be solved are given by $\hat x_i(t) = x_i(t) -\delta_{t\in\N}\cdot D_i(t)$. I.e., these are all tasks available, minus those that are going to be redistributed. These $\hat x_i(t)$ tasks are solved at rate $\tau_i$.
The remaining terms remove redistributed tasks and add new tasks and other agents tasks to the pool of tasks to be dealt with.
Hence, eq 7 takes the following form:
\frac{d x_i(t)}{dt} =& -\tau [x_i(t) -\delta_{t\in\N}\cdot D_i(t)] - \delta_{t\in\N}\cdot D_i(t) + \delta_{t\in\N}\cdot \sum_j d_{ji}(t) + \delta_{t \text{ mod} T_\text{new} = 0}
We have corrected the manuscript accordingly.
4) The distribution of fitness, $\phi_i$, is a Gaussian with $\mu=50$ and different standard deviations $\sigma$ (line 59, page 2): that means that there is always the possibility to assign negative values to the fitness. Now, for small values of $\sigma$ and small systems this probability is more than negligible, but, for instance, with $\sigma=22$ and $N=100$ (values utilized in Figure 3), it implies that on average 1.3 agents have a negative fitness for each realization, so that we probably in some realizations there were several agents with negative fitness. I do not think this changes apreciably the results obtained, but how did you treat these few cases? If negative, the fitness is like zero, but this means that the "real" distribution is not exactly a normal one. I understand that the practical effect is not important, but it would be better to clarify this point;
Agents with negative fitness or zero fitness are assumed to be failed from t=0 (see eq(1)).
Thus, simulations with large $\sigma$ on average have a non-zero proportion of failed agents at the start of the simulations.
We have included this explanation in the manuscript.
5) Figure 2, page 6: for the systems utilized, the maximum of $H(A$) is around 6, which implies that the number of agents is 20, but the system depicted are made up by 10 nodes (which would imply a maximum of $H(A)$ around 4). Are the networks depicted in the figure the real networks of the simulated systems?
The plotted curves and the networks in Figure 2 on page 6 have been simulated using 10 agents.
The maximum entropy values follow from Equation (9) in the paper:
H = 0.9 * log_2(100) \approx 6,
where we chose base 2 (as in bits, following the custom for Shannon information entropy) for the entropy units, and not base e.
Likely, the confusion comes up for this reason.
To address this issue, we now explicitly write "\log_2" instead of "\log" to be clear on the base we use for the logarithm.
Minor points:
6) Figure 3, page 7: the heatmap shows what is called the "probability" of $X(t)$ for given values of failed agents' ratio and $\sigma$, but I'd call it "frequency" (I do not mean "probability" is wrong, but since I presume it was measured just counting the realizations ended up with the given value of $X(t)$, "frequency" is in my opinion better);
In figure 3, we plot an estimated probability density (obtained from a standard kernel density estimation), and not a frequency of observations.
We have updated the figure caption to clarify this.
7) Throughout the paper, there are figures with "red" curves, nodes, etc., but it seems orange (unless there is some problem with the visualization in my laptop). Of course it is not really an issue but just to be precise.
Thanks for pointing out this issue. We had modified the colors in the figure without a corresponding change of the descriptions in the manuscript.
We have now updated blue --> violet, red --> orange to reflect the used colors.
Round 2
Reviewer 2 Report
The authors have fullfilled my requests and now I endorse the manuscript for publication.